# 1,2,4-Triazoles as Important Antibacterial Agents

**DOI:** 10.3390/ph14030224

**Published:** 2021-03-07

**Authors:** Małgorzata Strzelecka, Piotr Świątek

**Affiliations:** Department of Medicinal Chemistry, Faculty of Pharmacy, Wroclaw Medical University, Borowska 211, 50-556 Wroclaw, Poland; malgorzata.strzelecka@umed.wroc.pl

**Keywords:** 1,2,4-triazole, antimicrobial activity, antibacterial drugs

## Abstract

The global spread of drug resistance in bacteria requires new potent and safe antimicrobial agents. Compounds containing the 1,2,4-triazole ring in their structure are characterised by multidirectional biological activity. A large volume of research on triazole and their derivatives has been carried out, proving significant antibacterial activity of this heterocyclic core. This review is useful for further investigations on this scaffold to harness its optimum antibacterial potential. Moreover, rational design and development of the novel antibacterial agents incorporating 1,2,4-triazole can help in dealing with the escalating problems of microbial resistance.

## 1. Introduction

Since the discovery of the first antibiotic (Penicillin, 1928), there has been an ongoing ”race” between scientists developing new antibacterial agents and pathogenic bacteria harboring various resistance mechanisms. In 2017, the World Health Organization (WHO) published a list of 12 bacteria whose level of resistance to antibiotics is such that they represent a major concern to public health, and grouped them according to their priority as critical, high, and medium. Critical-priority bacteria included Gram-negative bacterial pathogens, namely carbapenem-resistant *Acinetobacter baumannii* and *Pseudomonas aeruginosa*, as well as carbapenem-resistant and third-generation cephalosporin-resistant *Enterobacteriaceae*. The highest ranked Gram-positive bacteria (high priority) were vancomycin-resistant *Enterococcus faecium* (VRE) and meticillin-resistant *Staphylococcus aureus* (MRSA). According to another report, entitled “Tackling Drug-Resistant Infections Globally: Final Report and Recommendations”, which was a result of an extensive work carried out by an independent commission, drug-resistant infections may be the cause of 10 million deaths annually by 2050, exceeding the number of deaths attributable to road traffic accidents and even cancer [1]. It is not surprising given strong selection pressure exerted on bacteria by antibiotics that are overused and misused in humans and animals as well as in agriculture and food chains, which has also led to considerable environmental pollution. To overcome the development of drug resistance, it is crucial to search for new antibacterial molecules with novel mechanism of action as well as structural modification or optimization of the existing agents by improving both the binding affinity and the spectrum of activity while retaining bioavailability and safety profiles. Searching for new therapeutic options in the treatment of resistant bacterial infections include discovering new synthetic compounds as well as naturally occurring substances (especially from essential oils extracted from plants) [2,3,4].

A wide variety of heterocyclic systems have been explored in order to develop pharmaceutically important molecules. Nitrogen-containing heterocycles are found in many medicines. Derivatives of triazole have particularly interesting therapeutic properties. Triazoles are five-membered rings, which contain two carbon and three nitrogen atoms, with a molecular formula of C_2_H_3_N_3_. According to the position of nitrogen atoms, triazoles exist in two isomeric forms - 1,2,3-triazole and 1,2,4-triazole (Figure 1). The 1,2,4-triazole ring may exist in equilibrium between two forms: 1*H*-form and 4*H*-form (Figure 1). The calculated energy differences between azole tautomers support preference for the 1*H* over 4*H* tautomer [5].

Literature review shows that 1,2,4-triazoles and their fused heterocyclic derivatives show a wide spectrum of biological activities. The 1,2,4-triazole core has been incorporated into a wide variety of therapeutically important agents available in clinical therapy, such as itraconazole, posaconazole, voriconazole (antifungal), ribavirin (antiviral), rizatriptan (antimigraine), alprazolam (anxiolytic), trazodone (antidepressant), letrozole and anastrozole (antitumoral) (Figure 2). In the last few decades, scientists have been paying considerable attention to the synthesis of 1,2,4-triazole derivatives showing such comprehensive biological activities as an antifungal [6,7] antitubercular [8], antioxidant [9], anticancer [10], anti-inflammatory [11], analgesic [12], antidiabetic [13], anticonvulsant [14], and anxiolytic [15] activity. With regard to the antifungal activity, triazole-based pharmacophore has replaced the previously widely used imidazole pharmacophore in systemically active azoles, due to lower toxicity and higher bioavailability of triazole derivatives as well as an increased specificity for fungal cytochrome p450 and a lower impact on human sterol synthesis [16].

Many studies of triazole compounds relates to their antimicrobial properties. This review article highlights recent work (from 2010) carried out on 1,2,4-triazoles with potent antibacterial properties. The work systematizes the compounds in terms of their chemical structure. Antimicrobial properties of triazole hybrids with quinolones, 4-amino-, 3-mercaptotriazoles and fused triazole derivatives are discussed. The method of selecting publications for the review is presented in the diagram below (Figure 3).

## 2. Antibacterial Activity of Derivatives of 1,2,4-Triazole

Newly synthesized 1,2,4-triazole compounds were tested for their in vitro growth inhibitory activity against standard Gram-positive and Gram-negative bacterial strains. It is also recommended to test newly obtained substances with potential antibacterial effect on drug-resistant strains (e.g., MRSA, VRE). Additionally, studies on antituberculosis activity were carried out for some compounds. Preliminary screening was performed with the use of the agar disc-diffusion method (cup and plate method), measuring the growth inhibition zones on agar plates. The inhibition zone of compounds was measured using a centimeter scale. In another method, antimicrobial activity was expressed as the lowest concentration that completely inhibits visible growth of a microorganism as detected by unaided eye (minimum inhibitory concentration—MIC), and values of minimal bactericidal concentration (MBC) were also determined for some substances.

### 2.1. Triazole Hybrides of Quinolone Antibacterial Agents

A large number of 1,2,4-triazole hybrids with (fluoro)quinolone drugs have been incorporated into therapeutically interesting drug candidates in light of their potent antimicrobial effect, especially against resistant bacterial strains [17]. Novel nalidixic acid-based 1,2,4-triazole-3-thione derivatives **1**–**2** (Figure 4) were synthesized by Aggarwal et al. (2011) and screened against Gram-positive (*S. aureus*, *B. subtilis*) and Gram-negative bacteria (*E. coli*, *K. pneumoniae*, *P. aeruginosa*). The azomethine derivatives **1a***–***g** were found to be highly active against *P. aeruginosa* with MIC of 16 µg/mL. Among triazolothiadiazoles, compound **2** with chloro-substituent at the 2-position on the phenyl ring showed maximum antimicrobial potency against all tested microorganisms with MICs of 16 µg/mL, compared to the standard drug, streptomycin (MIC: 2–15 µg/mL) [18].

Ceylan et al. (2016) prepared Mannich bases **3** (Figure 4) bearing several biologically active amines (Am) and evaluated their antibacterial activity against *E. coli*, *Y. pseudotuberculosis*, *P. aeruginosa*, *S. aureus*, *E. faecalis*, *B. cereus* and *Mycobacterium smegmatis*. The results of the antimicrobial screening showed that compounds containing norfloxacine **3a**, ciprofloxacine **3b** or 7-aminocephalosporanic acid system **3c** showed excellent bacterial inhibition, especially against *Mycobacterium smegmatis,* with MIC values <1.9 μg/mL, which is better than that in the case of the standard drug, streptomycin (MIC = 4 μg/mL) [19].

In 2018, Mohamed et al. prepared 3-(5-amino-(2*H*)-1,2,4-triazol-3-yl)- naphthyridinone derivatives **4**–**5** (Figure 4). Results of antibacterial screening against a panel of bacteria revealed that both 3-(5-acylamino triazolyl)- **4** and 3-(5-benzylidineamino triazolyl)-naphthyridones **5** showed remarkable selectivity against *Bacillus subtilis* which was resistant to nalidixic acid. Furthermore, the introduction of bromine at *C*-6 of the naphthyridinone skeleton enhanced antibacterial activity. DNA-gyrase inhibition assay of the most active compounds showed that they possessed a moderate to high inhibitory effect. Results of molecular docking revealed that 1,2,4-triazole is a bioisostere to COOH group, and the presence of amide and imine groups introduced additional binding interactions with the DNA–gyrase cleavage complex with lower binding energy.[20].

In the group of 1,2,4-triazole-quinolone hybrids, work was also carried out on 6-fluoro analogues (fluoroquinolones), namely norfloxacin, ciprofloxacin, ofloxacin or clinafloxacin. Plech et al. (2013, 2016) prepared new hybrids **6** (Figure 5) using Mannich reaction of 1,2,4-triazole-3-thione derivatives with ciprofloxacin and formaldehyde, and tested them against Gram-positive (MRSA, *S. aureus*, *S. epidermidis*, *B. subtilis*, *B. cereus*, *M. luteus*) and Gram-negative bacteria (*E. coli*, *P. mirabilis*, *P. aeruginosa*). The inhibitory effect of selected hybrids on planktonic and biofilm-forming cells of *Haemophilus influenzae* and *Haemophilus parainfluenzae* was also investigated. Most ciprofloxacin derivatives were found to possess antibacterial activity higher than the activity of ciprofloxacin, both against Gram-positive and Gram-negative species. Additionally, selected compounds revealed a distinct inhibitory effect against planktonic and biofilm-embedded cells of the *Haemophilus* spp. Compound **6a** exhibited the highest *anti*-MRSA activity with MIC values about 16- and 8-fold lower than in the case of ciprofloxacin and vancomycin, respectively. It was observed that differences in the activity of ciprofloxacin-triazole hybrids resulted from the type of substituent at the *C*-3 position of the 1,2,4-triazole ring, and the most favorable antibacterial effect was obtained with a hydroxyphenyl fragment [21,22]. Therefore, in a subsequent study, Plech et al. (2015) synthesized compounds **7** (Figure 5) that differ in terms of the position of the hydroxyl group in the phenyl ring at the *C*-3 position and the structure of the substituent in the *N*-4 position of the 1,2,4-triazole core. Microbiological tests revealed that the newly obtained hybrids were, in the vast majority, much more potent than ciprofloxacin itself. Compounds **7a**, **7b** and **7c** showed most potent action against MRSA with MIC of 0.046 µM (MIC for **7c** = 0.045 µM) in comparison with vancomycin (MIC = 0.68 µM). Analysis of structure activity relationship (SAR) showed that changes in the position of the hydroxyl group did not significantly affect antimicrobial activity. The position *N*-4, which has secondary effect on antimicrobial activity, showed a major effect on the toxicity profile of the tested compounds. The presence of methylene linker between triazole and aryl substituent increased toxicity against human cells. Results of enzymatic studies carried out for the selected compounds showed that antibacterial activity of ciprofloxacin-triazole hybrids does not depend solely on the degree of their affinity to bacterial type II topoisomerases (DNA gyrase and topoisomerase IV) [23,24].

1,2,4-Triazole hybrids of 1-[(1*R*,2*S*)-2-fluorocyclopropyl]ciprofloxacin **8** (Figure 5) were synthesized and evaluated for their in vitro antibacterial activity against a broad panel of clinically important drug-sensitive and drug-resistant pathogens (methicillin-sensitive and methicillin-resistant *S. epidermidis*, methicillin-sensitive and methicillin-resistant *S. aureus*, *E. faecalis*, and *E. faecium* as Gram-positive bacteria, and *E. coli* and *E. coli* producting extended spectrum beta-lactamases (ESBLs), *K. pneumoniae* and *K. pneumoniae* ESBLs(+), *P. aeruginosa*, *A. coacetious*, *E. cloacae*, *E. aerogenes*, *S. marcescens*, *M. morganii*, *P. rettgeri*, *P. vulgaris*, *P. mirabilis*, *S. maltophilia*, *C. freundii* as Gram-negative bacteria) by Gao et al. (2018). Bioassay results revealed that all hybrids had great potency against the tested strains, especially Gram-negative ones, and antibacterial activity was more potent than in the case of the parent 1-[(1*R*,2*S*)-2-fluorocyclopropyl]ciprofloxacin, and was comparable to ciprofloxacin and levofloxacin against the majority of tested pathogens. The SAR analysis showed that compounds containing benzyl group in the 4-position of the 1,2,4-triazole inhibited the growth of Gram-positive bacteria more strongly than 4-phenyl derivatives [25].

Researchers from Turkey (2017) synthesized phenylpiperazine derivatives of 5-oxo- and 5-thioxo-1,2,4-triazole-fluoroquinolone hybrids **9** (Figure 5). Excellent activity against all tested pathogens (*E. coli*, *Y. pseudotuberculosis*, *P. aeruginosa*, *E. faecalis*, *S. aureus*, *B. cereus*, *M. smegmatis*) with MIC values ranging from 0.12 to 1.95 µg/mL were observed for all compounds [26]. In another study, Mermer et al. (2019) synthesized phenylpiperazine-triazole-fluoroquinolone hybrids **10** (Figure 5) that differ in the *N*-4 position of the 1,2,4-triazole core. Fluoroquinolone derivatives exhibited good to excellent activity against almost all microorganisms with MIC values ranging from 0.125 to 64 μg/mL. 1-Cyclopropyl-6-fluoro-4-oxo-7-[4-({(4-ethyl)/(4-phenyl)-3-[(4-phenylpiperazin-1-yl)-methyl]-5-tioxo-4,5-dihydro-1*H*-1,2,4-triazol-1-yl}methyl)piperazin-1-yl]-1,4-dihydro-quinolin-3-carboxylic acids **10a** and **10b** showed the highest antimicrobial activity against tested microorganisms. Furthermore, these hybrids displayed good DNA gyrase inhibition with IC_50_ values ranging from 0.134 to 1.84 μg/mL. To explain the mode of interaction between compounds and receptors, a molecular docking study was performed. With an average least binding energy of −9.5 kcal/mol, all compounds were found to have remarkable inhibitory potentials against DNA gyrase (*E. coli*) [27].

Antibacterial activity of new ciprofloxacin derivatives **11** (Figure 5) was also reported by Mohammed et al. (2019). Coupling of 5-aryl-4*H*-1,2,4-triazole-3-thione derivatives with acylated ciprofloxacin gave S-bridged hybrids. Biological screening results indicated that the *N*-allyl derivative with unsubstituted phenyl moiety at the *C*-5 position of triazole displayed the highest antimycobacterial activity, both against non-pathogenic *Mycobacterium smegmatis,* compared to the reference isoniazid (MICs: 3.25 µg/mL vs 5 µg/mL), and pathogenic drug-resistant and drug-susceptible strains of *Mycobacterium tuberculosis,* compared to levofloxacin and moxifloxacin (MICs: 4–32 μg/mL vs 0.03–8 μg/mL). Moreover, *N*-allyl compound displayed a broad spectrum of antibacterial activity against all bacterial strains tested (*S. aureus*, *K. pneumoniae*, *P. aeruginosa*, *E. coli*). From the docking studies, it was found that most potent compound showed additional binding interactions with the active site of *Mycobacterium* gyrase enzyme (extra hydrogen bond interaction between the triazole nitrogen *N*1 and the amino acid residues Arg D 182), which may explain its enhanced activity against *M. smegmatis* [28].

In 2015, novel tricyclic fluoroquinolones **12** (Figure 5) with a functional Mannich base moiety at the *C*-8 position of the fused system were synthesized and evaluated for their antimicrobial activity against Gram-positive *S. aureus* and MRSA, Gram-negative *E. coli* and multidrug-resistant *E. coli* (MDR *E. coli*) bacterial strains using ciprofloxacin as a standard. Compounds derived from aliphatic amines were less potent than those derived from heterocyclic amine donors. In particular, 2-methylpiperazine compound **12h** was highly active against MDR *E. coli* bacterial strain with MIC value of 0.25 µg/mL, about 30-fold more potent than ciprofloxacin [29].

An Indian research team (2012) designed and synthesized novel ofloxacin analogues **13** (Figure 6) as antimicrobial agents. It was observed that 1,2,4-triazole derivatives of ofloxacin with MIC values ranging from 0.25 to 1 µg/mL had antibacterial properties against all tested Gram-positive (*S. aureus*, *S. epidermis*, *B. subtilis*) and one Gram-negative (*E. coli*) pathogens, which were comparable to ofloxacin (MICs: 0.25–1 µg/mL) [30].

In 2012, Wang et al. synthesized a series of clinafloxacin-triazole hybrids **14** (Figure 7) and tested their activity against a panel of bacterial strains (*S. aureus*, *B. subtilis*, *M. luteus*, *E. coli*, *S. dysenteriae*, *P. aeruginosa*, *B. proteus*). Additionally, the research was also based on methicillin-resistant strain of *S. aureus*. Most of 1,2,4-triazole derivatives of clinafloxacin displayed high inhibitory efficacy towards both Gram-positive and Gram-negative bacteria with MIC values ranging from 0.25 to 32 µg/mL. Compounds **14a**, **14b** and **14c** with 4-toyl, 4-fluorophenyl and 2,4-difluorophenyl groups, respectively, showed the strongest activity against MRSA (MICs: 0.25 µg/mL), which were more active than chloramphenicol (MIC = 16 µg/mL) and their precursor clinafloxacin (MIC = 1 µg/mL) [31].

### 2.2. Antibacterial Activity of 4-Amino-1,2,4-Triazole Derivatives

Indian researchers (2010) reported a synthesis of 4-amino-5-aryl-4*H*-1,2,4-triazole derivatives **15** (Figure 8) and screened for in vitro antibacterial properties against *E. coli*, *B. subtilis*, *P. aeruginosa* and *P. fluoroscens* (recultured). Compound with 4-trichloromethyl group attached to the phenyl ring at the 3-position of triazole was observed to exhibit the highest antibacterial activity (MIC = 5 µg/mL and the zone of inhibition 14–22 mm), equivalent to ceftriaxone, while compounds containing 4-chloro and 4-bromo substituents showed good activity. Acetylation of the NH_2_ group at the 4-position and the presence of the free SH group at the 3-position of triazole caused a decrease in antibacterial activity against most bacterial strains [32].

Muthal et al. (2010) synthesized 5-substituted-3-pyridin-4-yl-1,2,4-triazoles **16** (Figure 8) as antibacterial and anti-inflammatory agents. In vitro assay indicated that compound with 4-hydroxyphenyl moiety inhibited the growth of all bacteria (*B. subtilis, S. aureus, P. mirabilis* and *S. typhi*) to the extent comparable to levofloxacin (zones of inhibition of the compound—26–27 mm, compared to 28 mm for levofloxacin). *Proteus mirabilis* was most sensitive to all tested compounds. Additionally, the selected compounds showed moderate anti-inflammatory activity in a carrageenan-induced rat paw oedema model [33].

In 2013, Gadegoni et al. reported synthesis and antibacterial activity of 3-[4-amino-3-(1*H*-indol-3-yl-methyl)-5-oxo(5-thioxo)-4,5-dihydro-1,2,4-triazol-1-yl]-propionitriles **17** (Figure 8) and their 1,3,4-oxadiazole analogues. All compounds were subjected to in vitro antimicrobial screening against *B. subtilis*, *S. aureus*, *M. luteus*, *P. vulgaris*, *S. typhimurium* and *E. coli*. Results revealed that triazoles were more potent than oxadiazole analogues, and the compound with thione-substituted triazole ring was equipotent with standard drug ampicillin against *B. subtilis*, *S. aureus* and *P. vulgaris* (MICs: 1.56, 1.56 and 6.25 µg/mL, respectively), while its 5-oxo analogue exhibited the strongest action against *E. coli* (MIC = 3.12 µg/mL). Additionally, all tested compounds exhibited considerable anti-inflammatory properties [34].

Researchers from China (2015) synthesized novel indole derivatives of 4-amino-4*H*-1,2,4-triazole-3-thiol with the use of the ultrasonic-assisted method, and determined in vitro antibacterial activity thereof against four pathogenic strains, namely, *E. coli*, *B. subtilis*, *P. aeruginosa*, and *S. aureus*. Preliminary results of antibacterial assays indicated that the amino-containing derivative **18** (Figure 9) exhibited potent inhibitory effect against *S. aureus* and *E. coli* (MICs: 2 and 8 µg/mL, respectively), which was 2- and 8-fold more potent than amoxicillin [35].

4-Amino-5-pyridin-3-yl-2,4-dihydro-3*H*-1,2,4-triazole-3-thione was prepared by Ceylan (2016) and converted to the corresponding Mannich **19** (Figure 9) and Schiff bases to assess their antimicrobial activity against three Gram-positive (*S. aureus*, *B. cereus*, *M. smegmatis*) and four Gram-negative bacteria (*E. coli*, *Y. pseudotuberculosis*, *P. aeruginosa*, *E. faecalis*). Mannich bases **19a**, **19c**–**d**, which contained 2-amino-4,6-dimethylpyrimidine, thiomorpholine, 1-(4-fluorophenyl)piperazine, and 6-aminopenicillanic acid moiety in triazole skeleton showed higher bacteriostatic activity against *Yersinia pseudotuberculosis* than the standard drug, ampicillin. Conversion to corresponding Schiff bases resulted in a decreased activity against most of bacterial strains [36].

A series of methylthio-linked pyrimidinyl-1,2,4-triazoles **20** (Figure 9) were prepared and screened for their antimicrobial activity by Sekhar et al. (2018). The results of the bioassay indicated that the tested compounds were more active against Gram-negative bacteria than Gram-positive ones. Compound **20f** containing 4-nitro substituent showed pronounced activity against *P. aeruginosa* in comparison with chloramphenicol [37].

Upmanyu et al. (2012) synthesized 4-(substituted acetylamino)-3-mercapto-5-(4-substituted phenyl)-1,2,4-triazole derivatives **21** (Figure 10) and tested them for their in vitro antibacterial activity against four bacterial strains (*S. aureus*, *B. subtilis*, *P. aeruginosa* and *E. coli*). The SAR analysis of the compounds indicated that 4-methoxy phenyl group is preferable at the 5-position of the triazole ring compared to 4-methyl group. Moreover, antimicrobial activity of the compounds was enhanced with an increase in the number of the carbon atom group (at the *C*-2 of acetamido group) at position *N*-4 of the triazole ring, and decreased with branch chain substitution [38].

Patel et al. (2010, 2013) designed 3-(3-pyridyl)-5-(4-substituted phenyl)-4-[*N*-(substituted 1,3-benzothiazol-2-yl)amino]-4*H*-1,2,4-triazole derivatives **22** (Figure 10) as antibacterial and antituberculosis agents. Preliminary screening showed that among 4-methylphenyl derivatives **22a**, compounds containing 6-flouro and 6-methyl substituents at bezothiazole score exhibited activity against Gram-positive bacteria (*S. aureus* and *S. pyogenes*), which was equal or even higher than in the case of ampicillin, used as a standard, while compound with 6-nitro substituent showed pronounced activity against Gram-negative bacteria (*E. coli* and *P. aeruginosa*), which was 4- to 8-fold higher than in the case of the standard drug. Moreover, 4-methylphenyl **22a** and 4-chlorophenyl triazoles **22c** with 4-chloro substituent on the benzothiazole ring showed potent antitubercular activity [39,40].

In 2016, a team of Egyptian scientists studied antibacterial activity of a new series of Mannich bases **23**, chloromethyl **24** and sulfide derivatives **25** (Figure 10) of 4-anilino-5-phenyl-4*H*-1,2,4-triazole-3-thiol against one Gram-positive bacterium (*B. cereus*) and two Gram-negative bacteria (*P. aeruginosa* and *E. coli*). Mannich bases **23** containing morpholine and piperidine moieties exhibited potent inhibitory effect against all types of bacteria, while sulfides **25** showed good activity against *Bacillus cereus*. The *E. coli* strain was found to be the most sensitive to chloromethyl compound **24** [41].

### 2.3. Antibacterial Activity of Schiff Bases of 4-Amino-1,2,4-Triazole Derivatives

Schiff and Mannich bases of 5-(1-adamantyl)-4-amino-2,4-dihydro-3*H*-1,2,4-triazol-3-thione **26** (Figure 11) were screened by Al-Omar et al. (2010) for their antibacterial activity against Gram-positive (*S. aureus*, *B. subtilis*, *M. luteus*) and Gram-negative bacteria (*E. coli*, *P. aeruginosa*). The results of the diameter of growth inhibition zones identified most antibacterial active compounds, which were then examined by minimum inhibitory concentration (MIC). 5-(1-Adamantyl)-4-[(4-hydroxybenzylidene)amino]-3-mercapto-1,2,4-triazole **26a** and two 4-ethoxycarbonyl-1-piperidyl Mannich bases **26b**–**c** showed most potent inhibitory effect (MICs: 1–2 µg/mL) against *Staphylococcus aureus* and *Bacillus subtilis,* comparable to gentamycin and ampicillin (MICs: 0.5–2 µg/mL) [42].

Another series of Schiff bases of 4-amino-5-(1-phenylethyl)-2,4-dihydro-3*H*-1,2,4-triazole-3-thione was prepared by Çalișir et al. (2010) as antibacterial agents against six human pathogenic bacteria (*S. aureus, S. epidermidis*, *E. coli*, *K. pneumoniae*, *P. aeruginosa*, *P. mirabilis*). The obtained data revealed that Schiff base **27** (Figure 11) with 4-nitrophenyl substituent showed antibacterial activity at 9 μg/mL against *S. epidermidis*, with the same MIC as in the case of cefuroxime, used as a standard drug [43].

Murthy et al. (2012) evaluated a series of Schiff and Mannich bases of isatin derivatives of 4-amino-5-benzyl-2,4-dihydro-3*H*-1,2,4-triazole-3-thione **28**–**29** (Figure 11) for their antibacterial activities against *S. aures*, *B. subtilis*, *P. aeruginosa*, and *E. coli*. The SAR indicated that compounds with chloro and bromo groups at the *C*-5 position of isatin exhibited broad-spectrum antibacterial activity with the inhibition zone of 20–27 mm, comparable to ciprofloxacin (inhibition zone of 25–30 mm). In general, antimicrobial activity of Mannich bases **29** was less than that of Schiff bases **28** [44].

In 2013, results of research on Schiff bases of *N*-[(4-amino-5-sulfanyl-4*H*-1,2,4-triazol-3-yl)methyl]-4-substituted benzamides **30** (Figure 11) as antibacterial agents were published by Mange et al. All newly synthesized compounds were evaluated for their antimicrobial activity against Gram-positive (*S. aureus* and *B. subtilis*) and Gram-negative bacteria (*E. coli* and *P. aeruginosa*). The authors observed that all compounds exhibited the same effect against *Staphylococcus aureus* as the standard drug, ceftriaxone, and moderate activity against other bacteria [45].

A group of scientists from China (2013) studied Schiff bases of symmetric disulfides connected to the 4-amino-3-(1-benzyl-1*H*-indol-3-yl]-5-thiomethyl-1,2,4-triazole **31** (Figure 12) for their antibacterial activity against *E. coli*, *S. aureus* and *P. aeruginosa*. They observed that a compound with 3-bromophenyl substituent showed strong activity against all three bacteria, equal to the reference, sparfloxacin. Compounds bearing an unsubstituted phenyl ring or 2-furyl showed poor activity against all bacterial strains [46]. In another study, the same researchers (2014) synthesized 3-[1-(4-fluorobenzyl)-1*H*-indol-3-yl]-5-(4-fluorobenzylthio)-4*H*-1,2,4-triazol-4-amine and its Schiff bases **32** (Figure 12). Results of antibacterial screening confirmed that the aromatic substituent at the 4-position of triazole played an important role in antibacterial activity and that the presence of halogen and nitro groups significantly enhanced inhibitory activity against all tested bacteria [47].

Ünver et al. (2016) developed a series of hybrid 1,2,4-triazole molecules **33**–**34** (Figure 13) derived from Schiff and Mannich bases, and evaluated them for antibacterial activity against a panel of bacteria, namely *E. coli*, *Y. pseudotuberculosis*, *P. aeruginosa*, *E. faecalis*, *S. aureus*, *B. cereus* and *Mycobacterium smegmatis*. Among the new compounds, the Schiff base **33d** carrying nitro substituent on the thiophene ring at the 4-position of 1,2,4-triazole showed the highest inhibitory activity against all tested species, 2- or even 35-fold higher than ampicillin. Compounds with morpholine **34** were generally less active [48].

Rajasekaran et al. (2017) designed Schiff bases of 1,2,4-triazole **35** (Figure 13) carrying various substituted aromatic groups at the *C*-3 and *N*-4 positions, and oxygen or sulphur at the 5-position of the triazole ring. All synthesized compounds were screened for their antibacterial activity against three Gram-positive (*M. luteus*, *S. albus* and *S. aureus*) and three Gram-negative bacteria (*E coli*, *P. aeruginosa* and *K. pneumoniae*). Two compounds, 4-[(4-hydroxy-3-methoxybenzylidene)amino]-5-(2-nitrophenyl)/(3-nitrophenyl)-2*H*-1,2,4-triazol-3-ones, showed 4-fold more potent antibacterial activity against *S. albus* than the standard drug, gatifloxacin (MIC = 12.5 µg/mL). The SAR analysis of Schiff bases revealed that the presence of 4-hydroxy-3-methoxyphenyl moiety at the *N*-4 position and the presence of the nitro group in the phenyl ring at the 3-position of 1,2,4-triazole played a crucial role in exerting high antibacterial activity against all species. Additionally, a docking study revealed that the new compounds had high affinity within the binding pocket of glucosamine-6-phosphate synthase, an enzyme involved in the assembly of the cell wall of microorganisms. Compounds having 4-hydroxy-3-methoxyphenyl group and the nitro group in the phenyl ring exhibited hydrogen bond interactions beetween oxygen of methoxy group (CH_3_O), oxygen and NH in triazolone ring, and oxygen of hydroxy group (HO) with three amino acid residues in the binding pocket of the enzyme, namely Ser347, Thr352 and Val399 [49].

4-Amino-5-(4-aminophenyl)-4*H*-1,2,4-triazole-3-thiol and its derivatives with functional Schiff-base moieties at the *N*-4 and *C*-5 positions of triazole were screened for their antimicrobial activity by Thakkar et al. (2017). The study revealed that 4-[(3-nitrobenzylidene)amino]-5-{4-[(3-nitrobenzylidene)amino]phenyl}-4*H*-1,2,4-triazole-3-thiol **36** (Figure 13) showed the highest activity against Gram-positive bacteria, namely *S. aureus* and *S. pyogenes,* with MIC values of 0.264 and 0.132 mM, respectively, which were equal to or higher than the activity of standard drugs, ampicillin and chloramphenicol. Moreover, a dihydrofolate reductase (DHFR) inhibition assay was conducted and the results thereof indicated that all compounds were potent DHFR inhibitors [50].

Antimicrobial activity of Schiff bases of thiazolyl-triazole hybrides **37** (Figure 13) was tested by Nastasa et al. (2018). The determination of inhibition zone diameters revealed that compounds **37a**–**b**, **37i**, and **37j** were most potent against Gram-positive *L. monocytogenes*, with an equal or greater effect than ciprofloxacin. MIC and MBC values were in line with the obtained results. With regard to the activity against Gram-negative strains, most of the compounds inhibited growth of *P. aeruginosa**,* with MIC and MBC values 2-fold more potent than the reference. A molecular docking study, performed on DNA-gyrase A and gyrase B from *L. monocytogenes*, revealed that all Schiff bases were stronger binders to gyrA than ciprofloxacin (used as the control inhibitor), and formed at least three hydrogen bonds between the azomethine nitrogen and serine (Ser98) and between the triazole nitrogens (*N*2 and *N*4) and the valine residues (Val113, Val268), while ciprofloxacin formed two hydrogen bonds between the carboxyl group from position 3 with Gly171 and Ser172 at the active site of enzyme. All compounds were considerably weaker binders to gyrB [51].

### 2.4. Antibacterial Activity of 1,2,4-Triazole-3-Thione Derivatives

Ali at al. (2010) synthesized 1,2,4-triazoles and 1,3-thiazines, incorporating acridine and 1,2,3,4-tetrahydroacridine moieties via heterocyclization of the key intermediate 4-(acridin-9-yl)-1-(1,2,3,4-tetrahydroacridin-9-ylcarbonyl)-thiosemicarbazide. 4-(9-Acridinyl)-5-(1,2,3,4-tetrahydroacridin-9-yl)-2,4-dihydro-3*H*-1,2,4-triazole-3-thione **38** (Figure 14) showed moderate inhibition against *Streptococcus pyogenes* and *Pseudomonas phaseolicola* at a concentration of 1 µg/mL (zone of inhibition 16–17 mm) and high inhibition at a concentration of 2 µg/mL (zone of inhibition 22–25 mm); they showed higher antibacterial activity against all microbial strains in comparison with the starting material [52].

Plech et al. (2011) prepared *N*-2-hydroxymethyl **39** and *N*-2-aminomethyl derivatives **40** (Figure 14) of 4-[4-bromophenyl(4-methylphenyl)]-5-(3-chlorophenyl)-2,4-dihydro-3*H*-1,2,4-triazole-3-thione to study an antibacterial activity of new compounds. Results of microbiological studies, carried out for six strains of Gram-positive and four strains of Gram-negative bacteria, revealed that the presence of the 4-bromophenyl moiety was crucial for a potent antibacterial effect in comparison with 4-methylphenyl derivatives. *Bacillus cereus* was the strain that was most sensitive to the new triazoles, and most compounds completely inhibited its growth at a concentration of 31.25–62.5 μg/mL [53]. In his subsequent research on triazoles, Plech et al. (2012) obtained Mannich bases from 4,5-disubstituted 1,2,4-triazole-3-thiones **40** (Figure 14) to check the impact of the substituent in the *C*-5 position on antibacterial activity. The obtained data indicated that the introduction of an electron-withdrawing chlorine atom to the phenyl ring in the *C*-5 position increased antibacterial potency against Gram-positive bacterial strains, namely *S. aureus*, MRSA, *S. epidermidis*, *B. subtilis*, *B. cereus* and *M. luteus*. None of the tested compounds inhibited the growth of Gram-negative bacteria. Derivatives of 4-(4-bromophenyl)-5-(4-chlorophenyl)-2,4-dihydro-3*H*-1,2,4-triazole-3-thione with pyrrolidinyl and diethylamino methyl substituents at the *N*-2 position were the most active against *Bacillus subtilis* (MICs: 31.25 μg/mL), which was comparable to cefuroxime, and 2-fold more potent than ampicillin [54].

In vitro antibacterial activity of 1,2,4-triazole-3-thiones **41** (Figure 14) with substituted piperazine against *S. aureus*, *B. subtilis*, *P. aeruginosa* and *P. mirabilis* was studied by a team from India (2011). The results of the study revealed that the presence of phenylpiperazine moiety was crucial for high antibacterial activity against all the microbial strains tested. Additionally, compounds with the phenyl ring at the N-4 position of triazole showed higher activity compared to triazoles substituted with alkyl and alkene groups. However, none of the compounds showed higher activity than norfloxacin, used as a reference drug [55].

A new series of 1,2,4-triazole-3-thiones **42** (Figure 14) with diarylsulfone moiety at the *C*-5 position and a substituted phenyl ring at the 4-position of triazole were screened for their antimicrobial activity against some bacteria (*S. aureus*, *B. cereus*, *E. coli*, *E. cloacae*, *A. baumannii* and *P. aeruginosa*) by Barbuceanu et al. (2012). The results revealed that compound **42d**, bearing bromo diphenylsulfone moiety and 3,4,5-trimethoxyphenyl fragment attached to the triazole ring, exhibited the strongest activity against *B. cereus*, with MIC value of 8 µg/mL [56].

In 2012, Zoumpoulakis et al. designed a synthesis of a series of sulfonamide 1,2,4-triazoles **43** (Figure 15) containing different aliphatic side chains at the nitrogen atom in the 4-position of the triazole ring. 5-[2-(*N,N*-Dimethylsulfamoyl)-4,5-dimethoxybenzyl]-4-*tert*-butyl-1,2,4-triazole-3-thione was observed to be most active against all tested species (Gram-negative bacteria: *E. coli*, *P. aeruginosa*, *S. typhimurium, E. cloacae* and Gram-positive bacteria*: B. cereus*, *M. flavus*, *L. monocytogenes*, *S. aureus*), with MIC and MBC values ranging from 0.24 to 0.48 µmol/mL, which was comparable or even higher than in the case of ampicillin, used as a reference drug (MIC and MBC values: 0.25–1.24 µmol/mL) [57].

In 2012, researchers from China synthesized a series of 1-(dihalobenzylo)-1,2,4-triazole-3-thiols, thioethers and thiones as well as some corresponding triazolium derivatives, and evaluated them for their antimicrobial activity against MRSA, *S. aureus*, *B. subtilis* and *M. luteus* as Gram-positive bacteria, as well as *E. coli*, *S. dysenteriae*, *P. aeruginosa* and *E. typhosa* as Gram-negative ones. In vitro screening revealed that 3,4-dichlorobenzyl triazole-3-thione **44** and triazoliums **45** (Figure 15) showed the most potent bioactivity against most tested bacteria. Especially hexyl triazolium **45a** exhibited comparable or even higher growth inhibitory efficacy for *S. aureus* (MIC = 2 μg/mL), *E. coli* (MIC = 1 μg/mL), *P. aeruginosa* (MIC = 4 μg/mL) and *E*. *typhosa* (MIC = 8 μg/mL) compared with the reference drugs, chloromycin (MICs: 1 μg/mL; 2 μg/mL; 16 μg/mL; 8 μg/mL, respectively) and norfloxacin (MICs: 0.5 μg/mL; 4 μg/mL; 1 μg/mL; 4 μg/mL, respectively) [58].

Koparir (2013) synthesized 5,5’-butane-1,4-diylbis[4-ethyl(4-allyl)-2,4-dihydro-3*H*-1,2,4-triazole-3-thione] and their Mannich bases **46**–**47** (Figure 15) and evaluated them with regard to *E. coli*, *S. aureus* and *P. aeruginosa* bacterial strains. Antibacterial screening revealed that compounds with 4-methyl piperidine (**46c**, **47c**) and trifluoromethyl phenyl piperazine moieties (**46g**, **47g**) showed excellent activity against all tested bacterial strains at concentrations 1.56–3.12 μg/mL, and a significant inhibition zone of 22–33 mm, compared to chloramphenicol (MICs: 3.12–6.25 μg/mL; zone of inhibition 16–30 mm) [59,60].

Antibacterial activity of 4-benzyl-5-(furan-2-yl)-2,4-dihydro-3*H*-1,2,4-triazole-3-thione and its Mannich bases **48** (Figure 15) was reported by Basoglu et al. (2013). Results revealed that the starting compound showed high activity against *E. coli*, *E. aerogenes*, and *Y. pseudotuberculosis* with the inhibition zone of 8–12 mm, while its Mannich bases with morpholine, phenyl piperazine or piperidine moieties exhibited additional activity against other tested microorganisms (*P. aeruginosa, S.* aureus, *E. faecalis*, *B. cereus*, and *M.*
*smegmatis*) with the inhibition zone of 6–20 mm. The activity of morpholine derivative against *E. faecalis* was equal to that of ampicillin [61].

5-(2-Aminothiazol-4-yl)-4-substituted-phenyl-4*H*-1,2,4-triazole-3-thioles **49a** and their acetylamine **49b** and thioureide derivatives **49c** (Figure 16) were designed, synthesized, and evaluated for their antimicrobial activity against a panel of Gram-positive (*S. aureus* and *B. subtilis*) and Gram-negative bacteria (*E. coli* and *P. aeruginosa*) by Hassan et al. (2013). Compound with a free 2-amino group and phenoxy moiety at the 4-position of the phenyl ring exhibited potent growth inhibition of all tested bacterial strains, comparable to gentamicin and ciprofloxacin. Diacetylation of the 2-amino-thiazole function (**49b**) produced moderately active compounds, similar to phenyl-thioureido analogues **49c** [62].

Al-Abdullah et al. (2014) synthesized Mannich bases of 5-(1-adamantyl)-4-substituted-1,2,4-triazol-3-thione **50** (Figure 16) and screened their antibacterial activity against a panel of bacterial strains (*S. aureus*; *B. subtilis*; *M. luteus*; *E. coli*; *P. aeruginosa*). Among the tested compounds, 5-(1-adamantyl)-4-phenyl-2-[4-(pyrid-2-yl)-piperazine-1-ylmethyl]-1,2,4-triazoline-3-thione **50a** and its 2-methoxyphenyl piperazine analogue **50b** showed excellent antibacterial activity with growth inhibition zones > 19 mm against all of the tested microorganisms, and were found to be more active than ampicillin and gentamicin [63].

4-[4-(1*H*-Benzo[*d*]imidazol-2-yl)phenyl]-5-benzyl-2*H*-1,2,4-triazole-3(4*H*)-thione **51b** (Figure 16) synthesized by Barot et al. (2017) was observed to exhibit a significant inhibitory effect against *Bacillus cereus* with MIC of 5 µg/mL, comparable to reference drugs, ofloxacin and metronidazole (MICs: 2–3 µg/mL). Its benzo[*d*]imidazolyl methyl analogue **51a** inhibited bacterial growth to a lesser extent [64].

An antimicrobial screening study of 4,5-disubstituted 1,2,4-triazole-3-thiones conducted by an Iranian research team in 2020 revealed that 5-(4-hydroxyphenyl)-4-(4-nitrophenyl)-2,4-dihydro-3*H*-1,2,4-triazole-3-thione **52** (Figure 16) displayed promising antibacterial activity against *Staphylococcus epidermidis* and *Acinetobacter baumannii*, with MIC values of 32 µg/mL (MIC values of gentamycin: 16 µg/mL and 1 µg/mL, respectively) [65].

### 2.5. Antibacterial Activity of S-Substituted 1,2,4-Triazole-3-Thione Derivatives

Several *S*-substituted 3-[(substituted benzylidene)amino]-1*H*-1,2,4-triazole-5-thione derivatives **53**–**54** (Figure 17) were synthesized and evaluated for antibacterial activity against *S. aureus* and *E. coli* by El-Feky et al. (2010). The study revealed that acetohydrazide compound, namely 2-{3-[(4-bromobenzylidene)amino]-1*H*-1,2,4-triazol-5-ylthio}-acetohydrazide was the most active against the tested bacteria with the inhibition zone of 26–21 mm, to the extent comparable to ampicillin (zone of inhibition: 31 mm for *S. aureus*, and 23 mm for *E. coli*) [66].

*N*-Cyclohexyl-2-[5-(4-pyridyl)-4-(p-tolyl)-4*H*-1,2,4-triazol-3-ylsulfanyl]-acetamide **55** (Figure 17) was evaluated for its antibacterial activity against *E. coli*, *S. aureus*, *P. aeruginosa* and *K. pneumoniae* by Orek et al. (2012). This compound showed good level of inhibition at a concentration of 6.25 µg/mL, comparable to ciprofloxacin, used as a standard drug [67].

Singh et al. (2013) synthesized asymmetric bis-1,2,4-triazoles **56** (Figure 17) and determined their in vitro antibacterial activity against *B. subtilus*, *S. aureus*, *E. coli*, and *P. aeruginosa*. Among the tested compounds, a series with S-methyl linker (*n* = 1), in particular 5-[(4,5-diphenyl-4*H*-1,2,4-triazol-3-ylthio)methyl]-4-phenyl(4-fluorophenyl)-4*H*-1,2,4-triazole-3-thioles, were observed to be most potent against all bacterial strains [68].

El Ashry et al. (2013) evaluated *S*-glycosides and *S*,*N*-diglycosides of 1,2-dihydro-5-(1*H*-indol-2-yl)-1,2,4-triazole-3-thione for antibacterial activity against two human pathogenic microbes. The obtained data revealed that *Bacillus subtilis* was far more sensitive to the newly obtained compounds than *Pseudomans aeruginosa*, and S-gycosylated 1,2,4-triazoles **57a**–**b** (Figure 17) showed the same antibacterial effect in comparison with the standard drug, chloramphenicol [69].

In vitro evaluation of antibacterial activity was carried out for *N*-benzothiazole derivatives of 2-[4-(naphthalen-1-yl)-5-(quinolin-6-yl)-4*H*-1,2,4-triazol-3-ylthio]acetamide **58** (Figure 18) by researchers from South Korea (2014). The SAR test revealed that the presence of an electron-releasing ethoxy substituent at the C-6 position on the benzothiazole ring was crucial for high activity against *P. aeruginosa*, providing an MIC value of 25 µg/mL, which was equivalent to that of ampicillin. Analogues with electron-withdrawing substituents, namely fluorine and bromine, exhibited potent action against Gram-positive bacteria, *Staphylococcus aureus* and *Bacillus cereus*, respectively, with half-fold potency compared to ampicillin [70].

Al-Abdullah et al. (2014) synthesized *S*-substituted derivatives of 5-(1-adamantyl)-4-phenyl-1,2,4-triazol-3-thiones **59** (Figure 18) and screened their antibacterial activity against a panel of bacterial strains. Among the tested *S*-substituted derivatives, optimal activity was shown by the 3,5-trifluormethyl benzyl analogue **59e**, being 4- to 8-fold higher against *S. aureus*, *B. subtilis*, and *P. aeruginosa* than in the case of ampicillin and gentamicin, and the activity against *M. luteus* and *E. coli* was comparable to the reference drugs [63].

Cui et al. (2016) designed a series of 1,2,4-triazole-pyrimidine derivatives linked by sulfur **60** (Figure 18), and then carried out extensive in vitro and in silico studies of their antimicrobial activity. Preliminary screening against two representative strains (*S. aureus* and *E. coli*) revealed two most potent compounds with 2-methyl- or 2-phenylthio moieties at the 2-position of the pyrimidine ring **60a**–**b**. Further studies of antibacterial activity against various strains of bacteria, including methicillin-resistant *S. aureus*, showed that these compounds significantly inhibited the growth of all bacteria (*E. coli*, *B. subtilis*, *B. anthracis*, and *S. aureus* isolated) with MIC values ranging from 0.8 to 5.2 µM, and were 10- and even ≥1600-fold more effective against MRSA than most clinically used antibiotics, namely ampicillin, polymyxin B, erythromycin, tetracycline, kanamycin, rifampicin, norfloxacin and even vancomycin. Moreover, compounds **60a**–**b** were highly effective against various MRSA strains with efflux pumps, indicating that they may be helpful in combating multi-drug resistance. Bioassays indicated that the tested 1,2,4-triazole-pyrimidine derivatives were potent inhibitors of SecA-dependent protein-conducting channel activity and protein translocation. The SecA protein is a widely conserved membrane protein, responsible for the secretion of virulence factors and directly accessible from the extracellular matrix. Therefore, SecA inhibitors have the potential for being developed as broad-spectrum antimicrobials and can overcome the effect of efflux pumps which are responsible for multidrug resistance [71].

### 2.6. Antibacterial Activity of Fused 1,2,4-Triazole Derivatives

Kumar et al. (2010) developed a new series of isopropylthiazole-derived 1,2,4-triazole moiety fused with 1,3,4-dihydrothiadiazole, 1,3,4-thiadiazole and 1,3,4-thiadiazine, and tested them as antibacterial agents. Antimicrobial study revealed that 1,2,4-triazolo[3,4-*b*] [1,3,4]thiadiazine **61** (Figure 19) demonstrated excellent activity against all tested Gram-positive and Gram-negative pathogens (*S. aureus*, *S. faecalis*, *B. subtilis*, *K. pneumoniae*, *E. coli* and *P. aeruginosa*) with MIC values ranging from 4 to 16 µg/mL (MICs of ciprofloxacin and norfloxacin ≤ 5 µg/mL). Moreover, in antituberculosis screening of these series, compound **61** exhibited potent activity against *Mycobacterium tuberculosis* H37Rv at MIC of 4 µg/mL, when compared with first line drugs, such as isoniazid (MIC = 0.25 µg/mL) [72].

In 2011, Badr et al. studied a series of fused 1,2,4-triazoles **62** (Figure 19) starting from 4-amino-5-(5-nitrofuran-2-yl)-4*H*-1,2,4-triazole-3-thiol. Antimicrobial activity against two bacterial strains, namely *S. aureus* and *E. coli*, was determined as the minimum inhibitory concentration (MIC) and the minimum bactericidal concentration (MBC). Compounds, 3-(5-nitrofuran-2-yl)-*N*-aryl-1,2,4-triazolo[3,4-*b*][1,3,4]thiadiazol-6-amines **62a**–**c** inhibited the growth of *S. aureus* in a concentration of 25 µg/mL, which was equivalent to that of standard ampicillin [73].

Synthesis and antimicrobial evaluation of novel 3-aryl-6-arylamino-1,2,4-triazolo[3,4-*b*][1,3,4]thiadiazoles **63** (Figure 19) were conducted by Plech et al. in 2012. The analyzed compounds showed selective activity against Gram-positive bacteria, both against drug-sensitive (*S. aureus*, *S. epidermidis*, *B. subtilis*, *B. cereus*, *M. luteus*) and drug-resistant (MRSA) strains. The SAR analysis revealed that the presence of chlorine atom at the *meta*-position of the phenyl ring at *C*-3 of the 1,2,4-triazolo[3,4-*b*][1,3,4]thiadiazole system was the most beneficial for the *anti*-MSSA and *anti-*MRSA activity. The role of substituents at the 6-position is secondary. 3-Chlorophenyl derivatives **63e**–**f** were observed to have the highest *anti*-MRSA activity, comparable to the activity of vancomycin (MIC = 0.98 µg/mL) [74].

In 2012, results concerning antimicrobial activity of 3-(aryloxymethyl)-6-aryl(6-aryloxymethyl)-1,2,4-triazolo[3,4-*b*][1,3,4]thiadiazole derivatives **64** (Figure 20) against Gram-positive (*S. aureus*, *B. subtilis*), Gram-negative bacteria (*P. aeruginosa*, *E. coli*), and *Mycobacterium tuberculosis* H37Rv were published. The results indicated that compounds with 2,4-dichlorophenoxy methyl group at the *C*-6 position showed comparable anti-tubercular activity at MIC of 0.50 µg/mL, when compared with standard drugs, rifampicin and isoniazid (MICs: 0.25 µg/mL). Moreover, 6-[(2,4-dichlorophenoxy)methyl]-3-[(4-chloro-phenoxy)methyl]-1,2,4-triazolo[3,4-*b*][1,3,4]thiadiazole exhibited the highest activity against all bacterial strains (MICs: 1–2 µg/mL), compared with ceftriaxone (MIC = 1 µg/mL), used as a reference drug [75].

Several simple or aryl hydrazone pyrazole-containing derivatives of 1,2,4-triazolo[3,4-*b*][1,3,4]thiadiazin-6-yl-2*H*-pyran-2-one were obtained by Penta et al. (2013) via one-pot, multicomponent reaction. The results of antibacterial activity against Gram-positive (*S. aureus* and *B. subtilis*) and Gram-negative bacteria (*E. coli* and *K. pneumoniae*) showed that some aryl hydrazone derivatives inhibited the growth of all bacterial strains. Among them, compound containing a 3-nitro group on the phenyl ring **65** (Figure 20) showed a maximum zone of inhibition (24.0–35.0 mm), even greater than that of the standard drug, kanamycin (zone of inhibition 23.0–32.0 mm). In the MIC tests, this compound was identified as the most potent inhibitor with significant MIC values ranging from 6.0 to 10.0 µg/mL, which was more or equally potent as kanamycin (MICs: 4.0–11.0 µg/mL) [76].

In 2013, Behalo et al. synthesized a series of fused 1,2,4-triazole derivatives **66**–**67** (Figure 20), starting with 4-amino-5-[(quinolin-8-yloxy)methyl]-4*H*-1,2,4-triazole-3-thiole to determine in vitro antibacterial activity against *S. aureus* and *E.coli*. The tested compounds showed a moderate **66a**–**b** (inhibition zone of 13–15 mm) to high effect **67b** (inhibition zone of 20–21 mm) against the tested bacteria, in comparison with tetracycline. It is interesting, that 3-((quinolin-8-yloxy)methyl)-1,2,4triazolo[3,4-*b*][1,3,4]thiadiazol-6(5*H*)-thione **67b** inhibited bacterial growth to a greater extent than its 6-oxo analogue **67a**, which showed no antibacterial effect [77].

Reddy et al. (2015) synthesized 1,2,4-triazolo[3,4-*b*][1,3,4]thiadiazines **68** (Figure 21) containing 5-methyl-1-phenyl-1*H*-4-pyrazolyl moiety at the 3-position of triazole, and evaluated their antibacterial activity against four human pathogenic bacteria (*E. coli*, *K. pneumoniae*, *S. dysentriae*, and *S. flexnei*). The results revealed that fused compounds with substituted the phenyl ring attached to the thiadiazine core exhibited high antibacterial activity compared to an unsubstituted derivative. The zone of inhibition of compound **68b** at a concentration of 100 µg/mL was greater than that of neomycin, and almost equal to streptomycin, used as standards [78].

In 2017, a series of 1,2,4-triazolo[3,4-*b*][1,3,4]thiadiazole derivatives **69** (Figure 21) containing thiouracil moiety were synthesized by structural modifications on a known SecA ATPase inhibitor by Cui et al. All the compounds were evaluated for their antibacterial activity against *Bacillus amyloliquefaciens*, *Staphylococcus aureus*, and *Bacillus subtilis* (expressed as inhibitory rate/% at 25 µg/mL), and the results showed that two compounds containing 2,4-dichlorophenyl group attached to thiouracil moiety exhibited the strongest antibacterial activity against *B. subtilis*, with inhibitory rate above 90%, higher than that of norfloxacin and the known SecA inhibitor. The SAR analysis suggested that the introduction of additional chlorine atoms on marginal phenyls was beneficial for antibacterial activity [79].

In 2019, an Indian research team synthesized a series of phenylquinoline-1,2,4-triazolo[3,4*-b*][1,3,4]thiadiazines **70** (Figure 21) and evaluated their in vitro antimicrobial activity against *S. aureus*, *E. coli*, *P. aeruginosa*. The MIC and zone of inhibition data revealed that compounds with a substitution of the halogen atom at the 6-position of the phenylquinoline ring showed the highest antibacterial activity (MICs: 1–8 µg/mL), comparable to standard ampicillin (MICs: 1–4 µg/mL). Replacement of halogens with electron-donating groups (methyl, nitro, methoxy and *tri*-methoxy groups) clearly reduced antibacterial potentials [80].

A series of imidazo[2,1-*c*][1,2,4]triazoles were synthesized in multicomponent reaction, and evaluated for in vitro antimicrobial potential against *B. cereus*, *S. aureus*, *E. coli*, *P. aeruginosa* and *Salmonella enteritidis* (food isolate) by Aouali et al. (2015). Among them, *para*-chloro substituted compound **71** (Figure 22) emerged as a promising antibacterial agent against *B. cereus* and *S. aureus*, respectively, with the diameter of the inhibition zone ranging from 29 to 20 mm, and MIC values of 0.078 µg/mL and 0.312 µg/mL. Gram-negative bacteria were resistant to the tested compounds [81].

Schiff bases, amides, imide as well as thiourea derivatives of 7-amino-3-phenyl-1,2,4-triazolo[4,3*-a*]pyrimidin-5-(1*H*)-one were synthesized by Abu-Hashem et al. (2017) to screen their antibacterial activity against *S. aureus*, *B. subtilis*, *P. aeruginosa*, *E. coli*, and *S. typhi*. The inhibitory activity of the compounds against Gram-positive bacteria was generally higher than that of Gram-negative ones, and *P. aeruginosa* was the least sensitive strain. Compounds, 2-amino-*N*-(1,5-dihydro-5-oxo-3-phenyl-[1,2,4]triazolo[4,3-*a*]pyrimidin-7-yl)benzo[*b*]thiophene-3-carboxamide **72**, and 1-benzyl-, 1-ethyl- and 1-cyclohexyl thiourea derivatives **73b**–**d** (Figure 22) were most potent against *S. aureus* (inhibition zone of 26.60–30.20 mm) and *B. subtillis* (inhibition zone of 28.60–33.80 mm), when compared with penicillin G and streptomycin (inhibition zones of 30.50–34.25 and 28.00–30.00, respectively) [82].

Krishna et al. (2014) examined in vitro antimycobacterial (*Mycobacterium tuberculosis* H37Rv) and antibacterial activities (*B. subtilis*, *S. aureus*, *E. coli*, *P. aeruginosa*) of 1,2,4-triazolo[5,1-*b*]quinazolin-9(3*H*)-ones **74** (Figure 22) containing diphenylamine moiety. Among them, two compounds with phenyl or 2-thiophenyl rings at the *C*-4 position of fused triazole exhibited significant antibacterial and antituberculosis activity at a concentration of 12.5 µM (MIC of ampicillin 6.25 µM; MIC of isoniazid >0.2 µM). Triazolo-quinazolinones with a substituted phenyl ring at the 4-position of molecule showed reduced activity when compared to unsubstituted derivative [83].

In 2011, two sets of novel dihydroindeno- and indeno[1,2-*e*][1,2,4]triazolo[3,4-*b*][1,3,4]thiadiazines **75**–**76** (Figure 23) were synthesized as structural analogs of 6-phenyl-7*H*-1,2,4-triazolo[3,4-*b*][1,3,4]thiadiazines and evaluated for in vitro antibacterial activity against four strains, namely, *S. aureus*, *B. subtilis*, *E. coli* and *P. aeruginosa*. The results revealed that Gram-positive bacteria were more sensitive to the inhibitory effect of the compounds. Furthermore, fused 10,10a-dihydroindeno derivatives **75** were more potent than their indeno analogues **76**. The most potent compounds, with MIC values ranging from 4 to 8 µg/mL (MIC of ciprofloxacin: 5 µg/mL), contained a methyl group at position 9 of the fused system [84].

A new class of fused (pyrazolo)/(isoxazolo)[3’,4’:4,5]thiazolo[3,2-*b*][1,2,4]-triazoles **77** (Figure 23) was prepared by Indian researchers (2016) in order to determine their antimicrobial properties. The results revealed that the tested compounds showed higher activity against Gram-positive bacteria (*Bacillus subtilis*, *Bacillus thuringiensis*) than Gram-negative bacteria (*Escherichia coli*, *Pseudomonas aeruginosa*). Moreover, compounds with electron-withdrawing substituents, such as chloro, bromo, nitro groups exhibited high activity against *M. tuberculosis* H37Rv with MIC values of 3.125 µg/mL, equal to streptomycin [85].

### 2.7. Miscellaneous 1,2,4-Triazoles with Antibacterial Activity

Antibacterial activity of bis-1,2,4-triazolium derivatives **78** (Figure 24) was reported by Thomas et al. (2019). The MIC values of the compounds were evaluated against four reference strains (*S. aureus*, *E. faecalis*, *E. coli* and *P. aeruginosa*), but also against four clinical isolates harboring various resistance mechanisms (MRSA, VRE, extended-spectrum b-lactamase-producing *Escherichia*, and *Pseudomonas aeruginosa* resistant, efflux pump). All the prepared bis-1,2,4-triazoliums showed strong activity against the majority of the tested strains, and the most active compound **78a** bearing decyl moiety was 2-, 4- and 8-fold more potent against the sensitive and the resistant strains of *S. aureus*, *E. faecalis* and *P. aeruginosa*, respectively, than the reference, chlorhexidine. Unfortunately, it also showed high toxicity [86].

In 2020, Stingaci et al. synthesized vinyl-1,2,4-triazole derivatives as antimicrobial agents. Compound **79** (Figure 24) exhibited excellent activity against all bacterial species (*B. subtilis*, *P. fluorescens*, *E. amylovora*, *E. carotovora*, *X. campestris*) with MIC and MBC ranging from 0.0002 to 0.0033 mM, which were comparable to ampicillin and chloramphenicol [87].

Researchers from Jordan (2020) prepared a series of 1,2,4-triazol-3-carbohydrazide derivatives **80** (Figure 24) and tested them against *S. aureus* and *B. cereus* as Gram-positive, and *P. aeruginosa* and *Shigella sp*. as Gram-negative bacteria. The results revealed that *Bacillus cereus* was the most sensitive bacterium, and compounds **80a**–**c** inhibited its growth equally to penicillin (inhibition zone of 10 mm). The calculated MIC values were in line with the obtained results [88].

### 2.8. Structure-Activity Observations

From the biological results, it becomes clear that different substituents on triazole scaffold have a noticeable effect on antibacterial activity. Making a general assessment of the relationship between the molecular structure and biological activity of the described compounds, it might be concluded that:

*I.* In the group of 1,2,4-triazole hybrides of quinolone, isosteric replacement of the COOH group with a 5-membered heterocyclic nucleus (among nalidixic acid and ofloxacine derivatives) or the incorporation of a differently substituted triazole moiety in the side chain at the *C*-7 position of fluoroquinolones (among nor-, cipro- and clinafloxacin derivatives) provides potent antibacterial properties. In particular, the presence of a hydroxyl group on the phenyl ring at the *C*-3 position of the triazole has improved antibacterial activity against the screened Gram-positive and Gram-negative bacterial strains. The formation of fused tricyclic fluoroquinolone-7-carboxylic acid derivatives by incorporating a triazole with a functional Mannich-based chain into the 7- and 8-positions of the fluoroquinolone scaffold greatly increased the antibacterial activity against drug-resistant bacteria.

*II.* Among the 4-amino-1,2,4-triazole derivatives, presence of a free –NH2 group and aryl substituents at the *C*-5 position of triazole provides a broad spectrum of antibacterial activity. The acetylation of an amino group or its replacement with aromatic amines, in particular 1,3-benzothiazol-2-amine, retains potent activity. The presence of an electron withdrawing group on the phenyl ring through -N=CH- linkage in the *N*-4 position of triazole (Schiff base derivatives), in many cases, is crucial for the high antibacterial activity.

*III.* The presence of aryl/heteroaryl substituent at *C*-5 position of 1,2,4-triazolo-3-thiones/thioles is crucial for potent antibacterial activity. In the group of 4,5-diphenyl-1,2,4-triazol-3-thione derivatives, the presence of an electron-withdrawing substituents at the phenyl rings enhanced the activity. The substitution of the N-2 position of triazole by various aminomethyl moieties (Mannich base derivatives) retains potent activity, and only after the introduction of large-volume substituents the efficiency of these derivatives decrease.

*IV.* In the group of triazoles fused with a 5- or 6-membered ring systems, namely 1,2,4-triazolo[3,4-*b*][1,3,4]thiadiazoles and 1,2,4-triazolo[3,4-*b*][1,3,4]thiadiazines, the aryl/heteroaryl substituents in the *C*-3 and *C*-6 positions of fused system have an impact on antibacterial activity, and with regard to the effect of the substituent on the phenyl ring among aryl derivatives, the most beneficial for the high antibacterial activity is the presence of halogen atom. The presence of large in volume substituents can decreases activity.

## 3. Summary

The conducted review of 1,2,4-triazole and their hybrids with quinolone agents as well as 4-amino-, 3-mercapto-, and fused derivatives of 1,2,4-triazole shows that they have potent antibacterial activity. These compounds inhibit the growth of both Gram-positive and Gram-negative bacteria and the most active compounds are equal or even more potent than the antibacterial drugs commonly used on the market. Moreover, some of 1,2,4-triazoles exhibit significant antibacterial activity against drug-resistant bacterial strains (e.g., MRSA, VRE, MDR *E. coli*) and antimycobacterial activity against *Mycobacterium tuberculosis.* The most active compounds are listed in Table 1. The study of the mechanism of action of some series of 1,2,4-triazole derivatives reveals that they have inhibitory potential against DNA gyrase, glucosamine-6-phosphate synthase, dihydrofolate reductase (DHFR) and SecA ATPase, which are essential proteins for bacteria. The structure-activity relationship (SAR) analysis provide the knowledge for further research and development of new 1,2,4-triazole derivatives with improved potency and maintained safety profile to overcome bacterial resistance.

## Figures and Tables

**Figure 1 pharmaceuticals-14-00224-f001:**
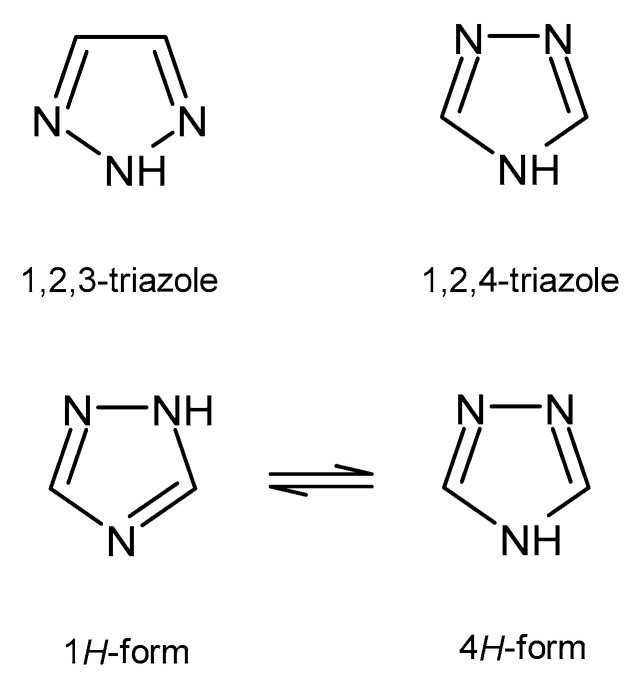
Two isomeric forms of triazole and tautomeric forms of 1,2,4-triazole.

**Figure 2 pharmaceuticals-14-00224-f002:**
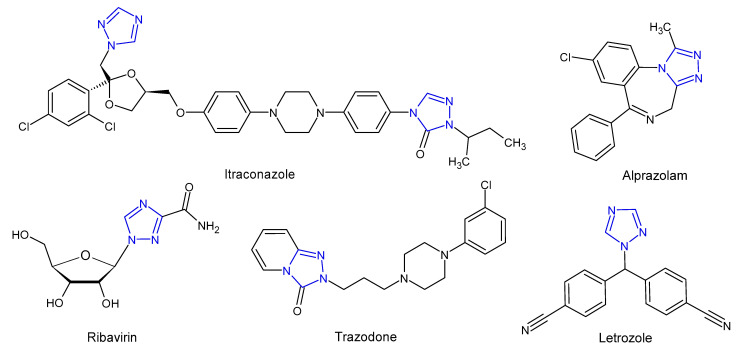
Selected 1,2,4-triazole drugs.

**Figure 3 pharmaceuticals-14-00224-f003:**
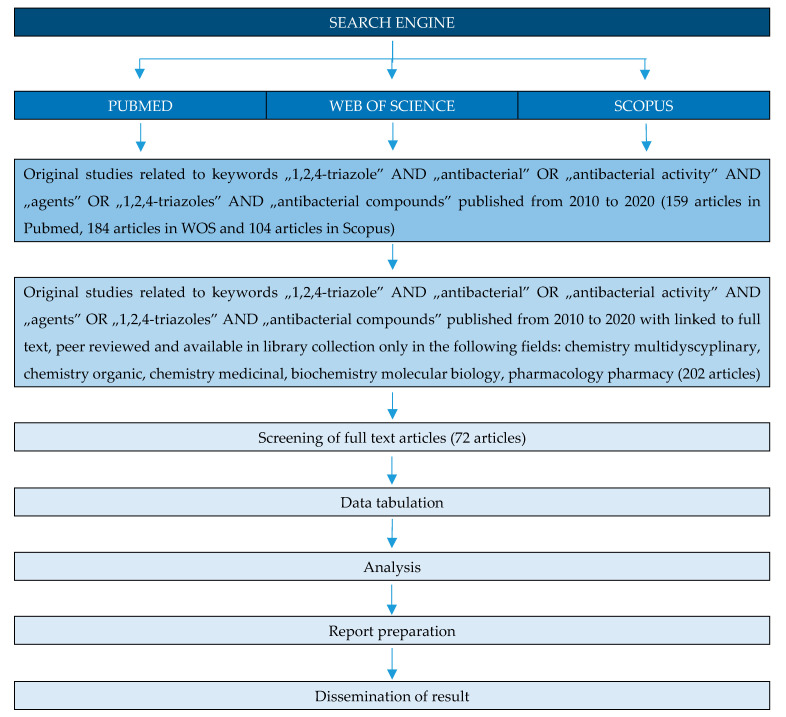
PRISMA flow diagram of the identification of literature for inclusion in this review.

**Figure 4 pharmaceuticals-14-00224-f004:**
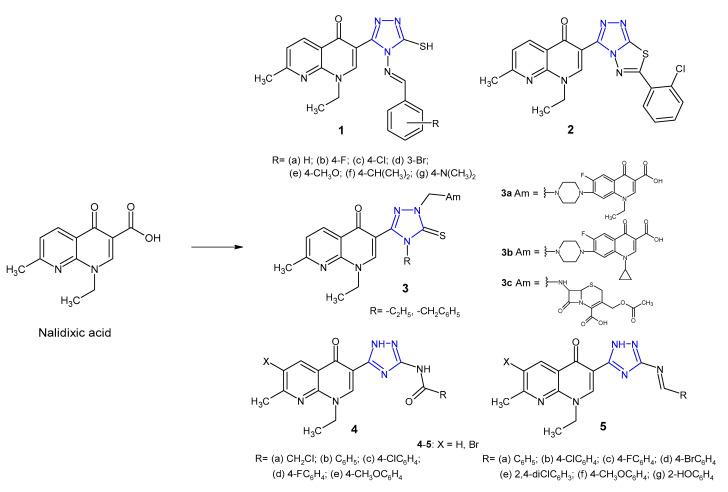
1,2,4-triazole analogues of nalidixic acid.

**Figure 5 pharmaceuticals-14-00224-f005:**
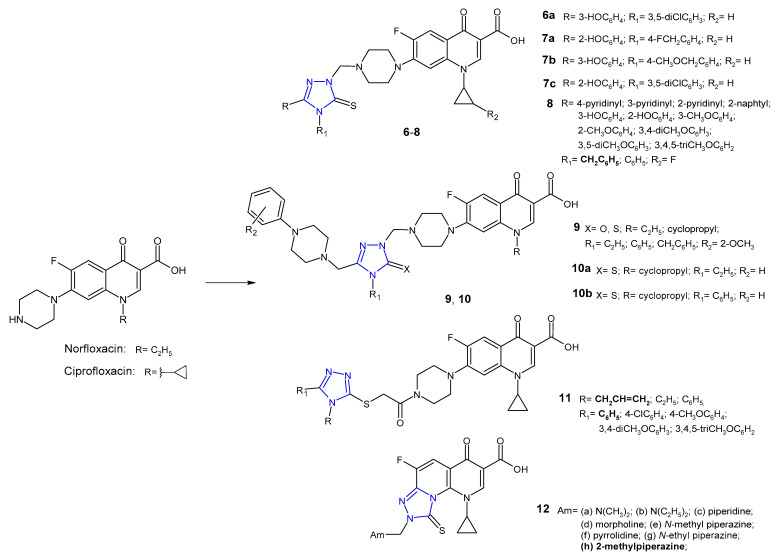
1,2,4-triazole analogues of norfloxacin and ciprofloxacin.

**Figure 6 pharmaceuticals-14-00224-f006:**
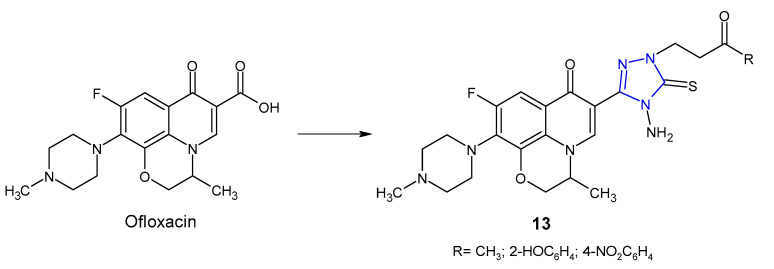
1,2,4-triazole analogues of ofloxacin.

**Figure 7 pharmaceuticals-14-00224-f007:**
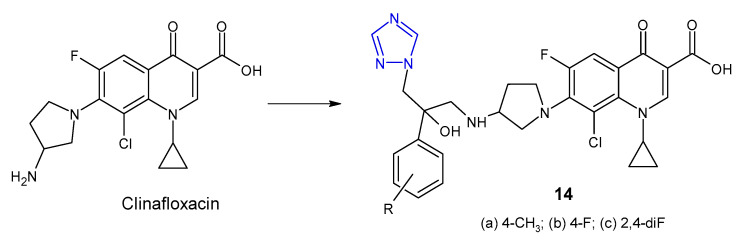
1,2,4-triazole analogues of clinafloxacin.

**Figure 8 pharmaceuticals-14-00224-f008:**
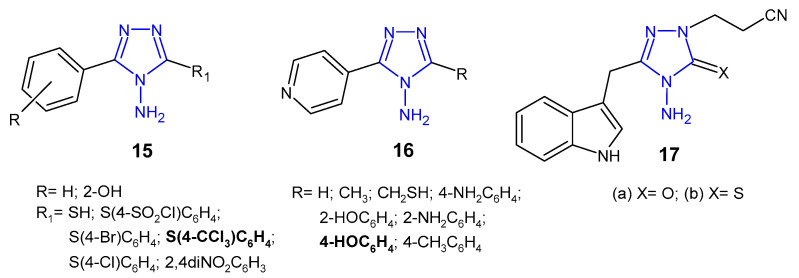
Antibacterial derivatives of 4-amino-1,2,4-triazole.

**Figure 9 pharmaceuticals-14-00224-f009:**
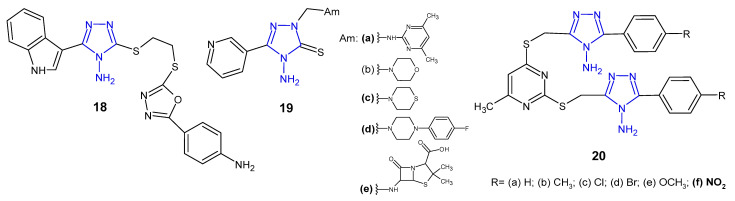
Antibacterial derivatives of 4-amino-1,2,4-triazole—continued.

**Figure 10 pharmaceuticals-14-00224-f010:**
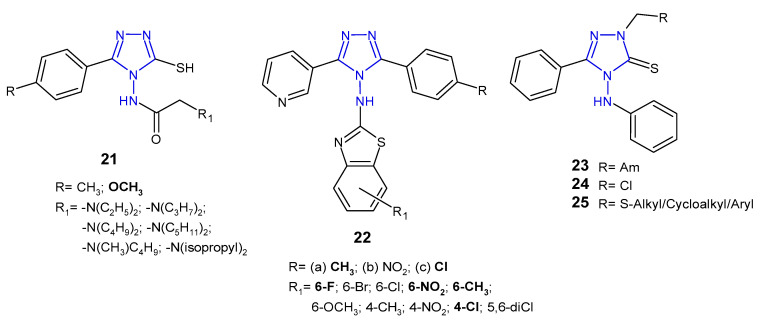
Antibacterial derivatives of acetylated and aromatic amines of 1,2,4-triazole.

**Figure 11 pharmaceuticals-14-00224-f011:**
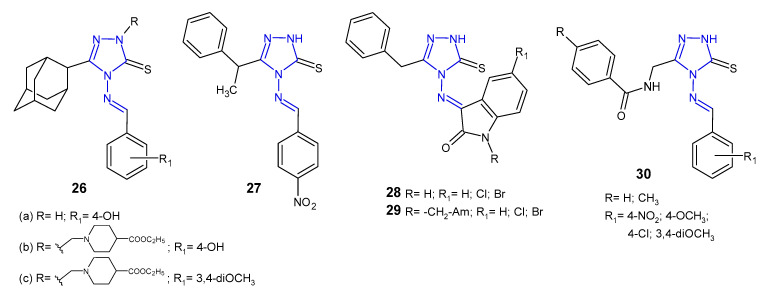
Antibacterial Schiff bases of 1,2,4-triazole.

**Figure 12 pharmaceuticals-14-00224-f012:**
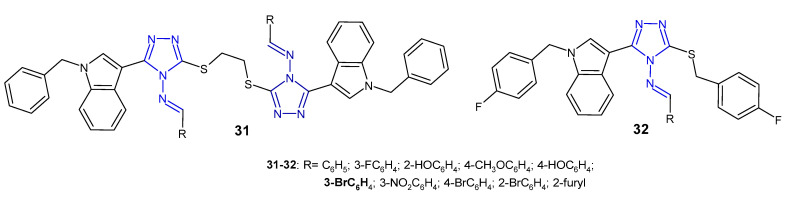
Antibacterial Schiff bases of 1,2,4-triazole—continued.

**Figure 13 pharmaceuticals-14-00224-f013:**
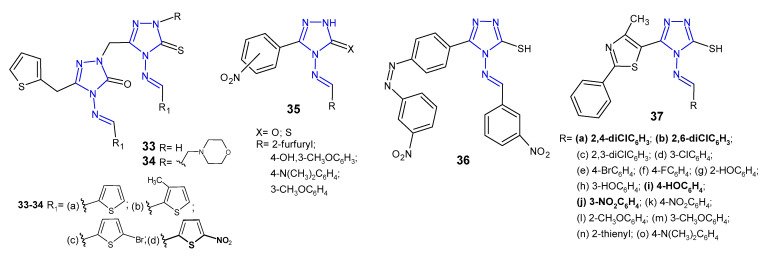
Antibacterial Schiff bases of 1,2,4-triazole—continued.

**Figure 14 pharmaceuticals-14-00224-f014:**
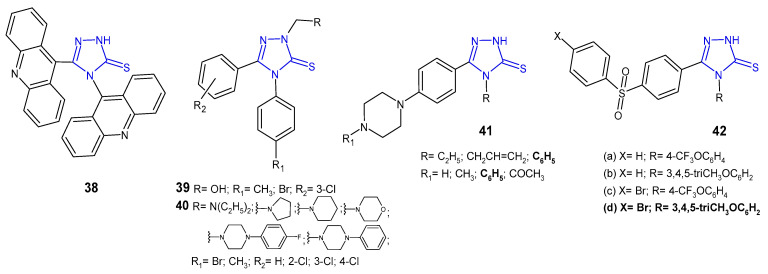
1,2,4-triazole-3-thiones as antibacterial agents.

**Figure 15 pharmaceuticals-14-00224-f015:**
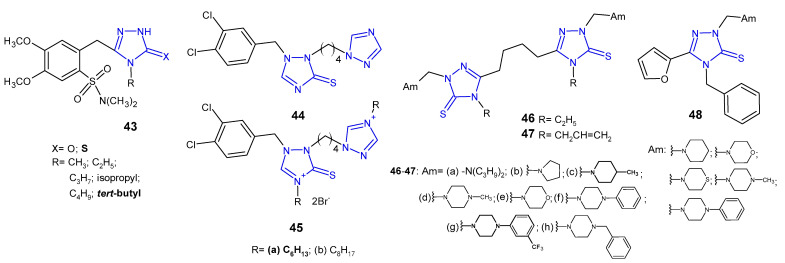
1,2,4-triazole-3-thiones as antibacterial agents—continued.

**Figure 16 pharmaceuticals-14-00224-f016:**
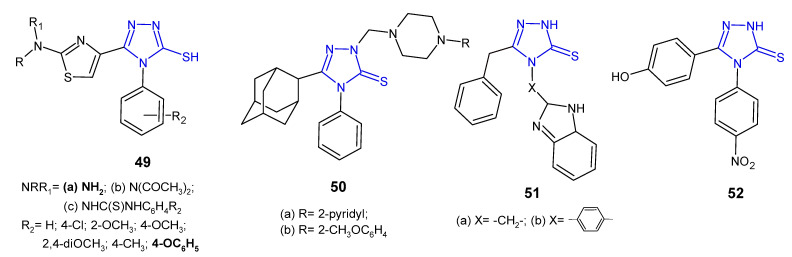
1,2,4-triazole-3-thiones as antibacterial agents—continued.

**Figure 17 pharmaceuticals-14-00224-f017:**
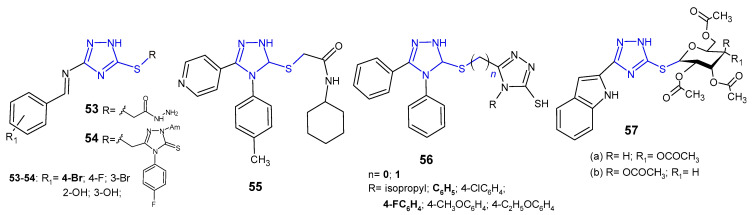
S-substituted 1,2,4-triazole-3-thioles with antibacterial activity.

**Figure 18 pharmaceuticals-14-00224-f018:**
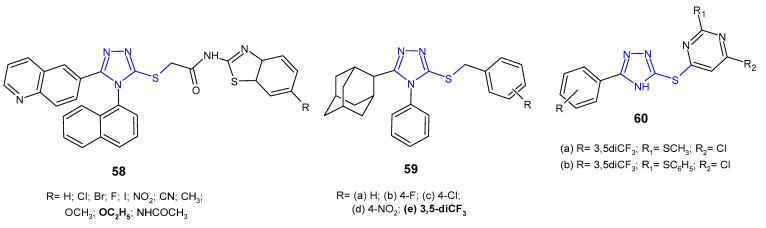
S-substituted 1,2,4-triazole-3-thioles with antibacterial activity-continued.

**Figure 19 pharmaceuticals-14-00224-f019:**
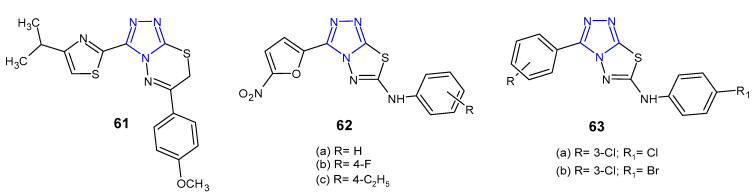
Antibacterial derivatives of 1,2,4-triazolo [3,4-b][1,3,4]thiadiazoles and 1,2,4-triazolo [3,4-b][1,3,4]thiadiazines.

**Figure 20 pharmaceuticals-14-00224-f020:**
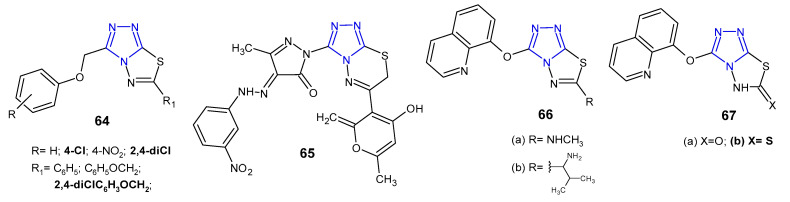
Antibacterial derivatives of 1,2,4-triazolo[3,4-*b*][1,3,4]thiadiazoles and 1,2,4-triazolo[3,4-*b*][1,3,4]thiadiazines—continued.

**Figure 21 pharmaceuticals-14-00224-f021:**
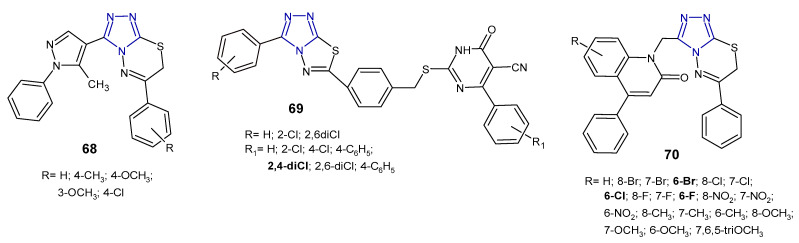
Antibacterial derivatives of 1,2,4-triazolo[3,4-*b*][1,3,4]thiadiazoles and 1,2,4-triazolo[3,4-*b*][1,3,4]thiadiazines—continued.

**Figure 22 pharmaceuticals-14-00224-f022:**
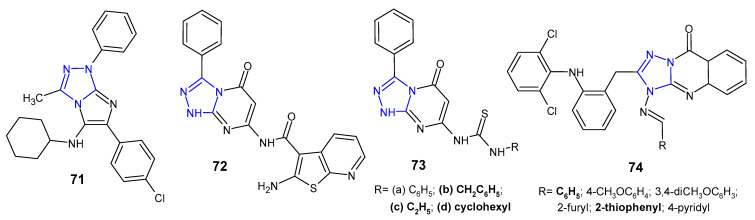
Another fused 1,2,4-triazole derivatives with antibacterial activity.

**Figure 23 pharmaceuticals-14-00224-f023:**
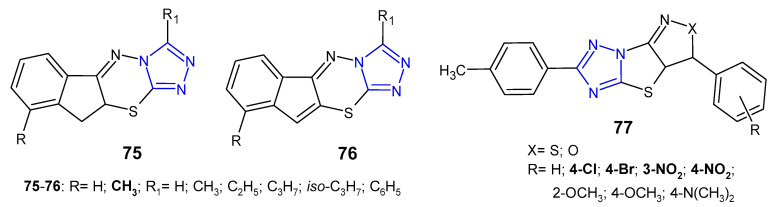
Another fused 1,2,4-triazole derivatives with antibacterial activity-continued.

**Figure 24 pharmaceuticals-14-00224-f024:**
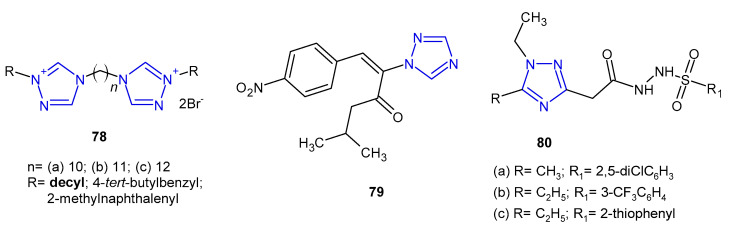
Miscellaneous 1,2,4-triazoles with antibacterial activity.

**Table 1 pharmaceuticals-14-00224-t001:** 1,2,4-triazole derivatives with the most potent antibacterial activity.

		Microorganisms and Minimal Inhibition Concentration (µg/mL)													
		Gram-Positive Bacteria							Gram-Negative Bacteria						
Compd.	Ref.	Drug-Susceptible					Drug-Resistant		Drug-Susceptible				Drug-Resistant		
		*S. aureus*	*S. epidermidis*	*B. subtilis*	*M. luteus*	Others	MRSA	VRE	*E.coli*	*P. aeruginosa*	Others		MDR *E.coli*	ESBLE	PAR
**6a** CPX VCN	[21]	0.72 2.96 -	0.35 1.48 -	< -	1.44 5.88 -	*B. cereus* 0.18 0.36 -	0.088 1.48 0.68	-	0.022 0.024 -	0.088 0.72 -	*P. mirabilis* 0.044 0.045 -	-	-	-	-
**7a**	[23]	0.19	0.046	0.011	0.76	*B. cereus* 0.093	0.046	-	0.011	0.046	*P. mirabilis* 0.023	*K. pneumoniae* 0.093			
**7b**		0.091	0.091	0.091	1.49	0.18	0.046		0.011	0.18	0.023	0.091	-	-	-
**7c** CPX VCN	[24]	0.09 0.72 -	0.18 1.48 -	0.09 0.09 -	0.70 5.88 -	0.09 0.36 -	0.045 – 0.68		0.024 0.024 -	0.176 0.72 -	0.044 0.045 -	0.088 0.36 -			
**12h** CPX	[29]	≤0.125 ≤0.25	-	-	-	-	≤0.5 ≤4.0	-	≤0.125 ≤0.25	-	-		≤0.25 ≤8.0	-	-
**14c** CFX CHL	[31]	0.5 0.5 16	-	0.5 0.5 32	0.5 0.5 8	-	0.25 1 16	-	1 0.5 32	0.5 0.5 32	*S. dysenteriae* 0.5 1 32	*B. proteus* 0.25 0.5 32	-	-	-
**18** AMX	[35]	2 16	-	<	-	-	-	-	8 16	<	-		-	-	-
**22a** (R_1_= 6-NO_2_)AMP	[39]	≤	-	-	-	-	-	-	12.5 100	25 100	-		-	-	-
**33d** AMP STR	[48]	0.98 35 -	-	-	-	*B. cereus* 0.98 15 -	-	-	3.9 10 -	31.25 >128 -	*Y. pseudotuberculosis* 7.8 18 -		-	-	-
**46c**,**g** **47c**,**g** CHL	[59] [60]	1.56 1.56 3.12	-	-	-	-	-	-	3.12 3.12 6.25	3.12 3.12 6.25	-		-	-	-
**59e** GEN AMP	[63]	0.5 2 2	-	0.5 2 0.5	2 2 2	-	-	-	2 0.5 2	<	-		-	-	-
**78a** (R= decyl) CHX	[86]	0.5 1	-	-	-	*E. faecalis* 0.5 2	0.5 1	0.5 2	0.5 0.5	2 8	-		-	1 1	2 16

[-] Not determined; [<] Antibacterial activity of compound is less than that in the case of the standard drug; Standard drugs: AMP—ampicillin, AMX—amoxicillin, CFX—clinafloxacin, CHL—chloramphenicol, CHX—chlorhexidine, CPX—ciprofloxacin, GFX—gatifloxacin, GEN—gentamicin, STR—streptomicin, VCN—vancomicin; Drug-resistant bacieria: MRSA—meticillin-resistant *S. aureus*, VRE—vancomycin-resistant *E. faecium*, MDR *E. coli*—multidrug-resistant *E. coli*, ESBLE—extended-spectrum b-lactamase-producing *E. coli*, PAR—*P. aeruginosa* resistant, efflux pump.

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
