# Peer review of "1,2,4-Triazoles as Important Antibacterial Agents"

_pharmaceuticals, 2021, doi:10.3390/ph14030224_

Round 1

Reviewer 1 Report

Strzelecka i ÅšwiÄ…tek descibed 1,2,4-triazole derivatives as important antibacterial agents. This review article highlights recent work (from 2010) carried out on 1,2,4-triazoles with potent antibacterial properties. In view of the phenomenon of increasing bacterial resistance, this topic is very import ant. The work is interesting, well planned and carefully written.

The work can be accepted for publication after minor corrections.

  1. Some Figures are too voluminous which makes reading difficult for example Figures 3, 4,7. I suggest dividing into smaller fragments.
  2. Why are some substituents bolded?
  3. MIC values should be harmonized (number of decimals). For example, line297.
  4. Figure 11- should be 4-HO-C6H4, Fig. 12 should be 3,4,5-triCH3OC6H2

Author Response

We sincerely appreciate your reviewing our contribution and giving us another opportunity to improve our manuscript in the best conceivable way. We considered all comments as constructive criticism. Considering the Reviewer comments, we have made a revision of our manuscript and our detailed answers are included below.

  1. Some Figures are too voluminous which makes reading difficult for example Figures 3, 4,7. I suggest dividing into smaller fragments.

We left Figures 3 and 4 unchanged because we wanted to emphasize that the compounds described in the manuscript are derivatives of nalidixic acid (Fig. 3) and norfloxacin and ciprofloxacin (Fig. 4). Figure 7 has been divided into two figures.

  1. Why are some substituents bolded?

Boldings refer to the substituents of the compounds which had the best activity in a given series. These compounds are described in more detail in the manuscript.

  1. MIC values should be harmonized (number of decimals). For example, line297.

The MIC values are presented as they appeared in the original publications. We didn't change it.

  1. Figure 11- should be 4-HO-C6H4, Fig. 12 should be 3,4,5-triCH3OC6H2

This has been checked and corrected on all figures in the whole manuscript.

Reviewer 2 Report

First of all, I appreciate the volume of work allocated to this paper. However, due to the article's style, it isn't easy to follow. The analyzed data are numerous, but the principles of systematization information are not very visible. 

Line 65-66. The purpose of the review is too general. I suggest to the authors to be more specific regarding this aspect.

The working methodology is missing (data sources, study selection, data extraction, etc.). The PRISMA statement and the associated website (www.prisma-statement.org/) should be helpful resources to improve your review.

The massive amount of collected data on antibacterial activity requires a table (or more, grouped by types of derivatives) to list the following columns: compound, method, tested bacterial species, MIC, the zone of inhibition, observations, and reference (and more others it is necessary). An ongoing challenge is the design of new molecules active on Gram-negative bacteria. Thus, it is useful to differentiate in the table the tested bacterial species in the two major categories, Gram-positive and Gram-negative.

In this manuscript, some presented studies provide details on the new compounds' synthesis, but others do not. Determine if this is an important issue. Consistency is required in writing the document from this point of view.

Also, assessments related to SAR appear only in the case of individual studies. From the SAR point of view, it would have been useful to identify the most valuable optimizations of the molecule that lead to increased antibacterial activity for each general group (starting to 2.1 to 2.5).

Docking studies are mentioned in the manuscript, which is a positive aspect. Molecular docking studies highlight the affinity of the new derivative to target enzymes. However, the references contain many more articles with molecular docking studies (at list ten). It would be interesting for readers to extract more relevant aspects related to the structural parts of the target compounds that have contributed to increasing the affinity for the target enzymes.

Author Response

Dear Reviewer,

We sincerely appreciate your reviewing our contribution and giving us another opportunity to improve our manuscript in the best conceivable way. We considered all comments as constructive criticism. Considering the Reviewer comments, we have made a revision of our manuscript and our detailed answers are included below.

  1. First of all, I appreciate the volume of work allocated to this paper. However, due to the article's style, it isn't easy to follow. The analyzed data are numerous, but the principles of systematization information are not very visible.

Due to the large amount of data, we decided to systematize the collected material in terms of the chemical structure of compounds. First, hybrids of triazoles with quinolones are discussed, then amino derivatives of triazoles, and then mercaptotriazoles, fused triazoles and triazole derivatives of different structure.

  1. Line 65-66. The purpose of the review is too general. I suggest to the authors to be more specific regarding this aspect.

The purpose of the work has been improved. The paper presents the results of literature review of triazole derivatives with antibacterial activity from the last 10 years.

  1. The working methodology is missing (data sources, study selection, data extraction, etc.). The PRISMA statement and the associated website (www.prisma-statement.org/) should be helpful resources to improve your review.

We got acquainted with the principles of writing systematic reviews and meta-analyzes on the PRISMA website. The publication was also prepared based on the guidelines of the MDPI editorial office.

  1. The massive amount of collected data on antibacterial activity requires a table (or more, grouped by types of derivatives) to list the following columns: compound, method, tested bacterial species, MIC, the zone of inhibition, observations, and reference (and more others it is necessary). An ongoing challenge is the design of new molecules active on Gram-negative bacteria. Thus, it is useful to differentiate in the table the tested bacterial species in the two major categories, Gram-positive and Gram-negative.

Thank you for this remark. Perhaps the tables would be more readable. In the manuscript, we very often discuss not individual compounds but a whole series. For this reason, we use general formulas in the figures. The most active compounds have the substituents in bold.

  1. In this manuscript, some presented studies provide details on the new compounds' synthesis, but others do not. Determine if this is an important issue. Consistency is required in writing the document from this point of view.

In the manuscript, we focused on the biological activity of triazole derivatives. We mention synthesis only when it is important for the antimicrobial activity of the compounds.

  1. Also, assessments related to SAR appear only in the case of individual studies. From the SAR point of view, it would have been useful to identify the most valuable optimizations of the molecule that lead to increased antibacterial activity for each general group (starting to 2.1 to 2.5).

We did not undertake to draw general conclusions about the relationship between activity and structure of compounds, because it is difficult to analyze the data when the authors of the source publications used different research models, various bacterial strains, and report their results using different values.

  1. Docking studies are mentioned in the manuscript, which is a positive aspect. Molecular docking studies highlight the affinity of the new derivative to target enzymes. However, the references contain many more articles with molecular docking studies (at list ten). It would be interesting for readers to extract more relevant aspects related to the structural parts of the target compounds that have contributed to increasing the affinity for the target enzymes.

New fragments of molecular docking were added to the manuscript, with particular emphasis on interactions between the compound and binding site, e.g. enzyme. We provide it only when the authors write about it in the source publication.

Reviewer 3 Report

Dear Authors

A large volume of research has been conducted on triazole and their derivatives, which have demonstrated a significant antibacterial activity of this heterocyclic nucleus. This review is useful for further investigation of this scaffold to exploit its optimal antibacterial potential. In addition, the rational design and development of novel antibacterial agents incorporating 1,2,4-triazole, also contained in phytocomplexes, can help address the growing problems of microbial resistance.

Introduction: must be reformed in the content and in the writing of the general part review the syntax of the topic

Discussion :  deepen the aspect concerning the use of natural molecules with antimicrobial activity against multi-resistant strains and infectious diseases. Learn more about this aspect using and citing the following references:

PMID: 32570731 ; PMID: 31454271 ; PMID: 27292570

 Check the bibliographic entries throughout the text, some of which are non-compliant, review some entries in the references and necessarily insert those referred to in point 2 for the purpose of acceptance by me.

  Best Regards

Author Response

Dear Reviewer,

We sincerely appreciate your reviewing our contribution and giving us another opportunity to improve our manuscript in the best conceivable way. We considered all comments as constructive criticism. Considering the Reviewer comments, we have made a revision of our manuscript and our detailed answers are included below.

Introduction: must be reformed in the content and in the writing of the general part review the syntax of the topic

The introduction has been improved. More detailed information on the purpose of the work was also introduced.

Discussion : deepen the aspect concerning the use of natural molecules with antimicrobial activity against multi-resistant strains and infectious diseases. Learn more about this aspect using and citing the following references:PMID: 32570731;PMID: 31454271; PMID: 27292570

According to the reviewer's suggestions, the introduction and the main part of the work have been corrected. Information on natural antibacterial compounds and antidiabetic triazole derivatives were introduced into the manuscript, and information on the activity of triazole-quinolone hybrids against Haemophilus was extended.

Check the bibliographic entries throughout the text, some of which are non-compliant, review some entries in the references and necessarily insert those referred to in point 2 for the purpose of acceptance by me.

Bibliographic entries throughout the text have been checked and corrected where necessary. The ones mentioned in point 2 have been inserted.

Round 2

Reviewer 2 Report

I appreciate the effort to improve the manuscript. However, some aspects that I mentioned previously were not met.

  1. The working methodology is still missing (data sources, study selection, data extraction, etc.). At least a brief description of it is required.
  2. To increase the value of the manuscript and to be more useful to readers, I firmly maintain my first recommendation: <The massive amount of collected data on antibacterial activity requires a table (or more, grouped by types of derivatives) to list the following columns: compound, method, tested bacterial species, MIC, the zone of inhibition, observations, and reference (and more others it is necessary). An ongoing challenge is the design of new molecules active on Gram-negative bacteria. Thus, it is useful to differentiate in the table the tested bacterial species in the two major categories, Gram-positive and Gram-negative.>
  3. The value of a manuscript increases through value of statements. From the SAR point of view, it is useful to identify the most valuable optimizations of the molecule that lead to increased antibacterial activity, as previously recommended.

Author Response

Dear Reviewer,

Thank you for your guidance. I hope that the attached, revised manuscript meet your expectations. Below my reply to your comments.

The working methodology is still missing (data sources, study selection, data extraction, etc.). At least a brief description of it is required.

We have added a diagram in which we presented the methodology for collecting and selecting source material. 

 To increase the value of the manuscript and to be more useful to readers, I firmly maintain my first recommendation: <The massive amount of collected data on antibacterial activity requires a table (or more, grouped by types of derivatives) to list the following columns: compound, method, tested bacterial species, MIC, the zone of inhibition, observations, and reference (and more others it is necessary). An ongoing challenge is the design of new molecules active on Gram-negative bacteria. Thus, it is useful to differentiate in the table the tested bacterial species in the two major categories, Gram-positive and Gram-negative.

We have added a table in which we have listed the most active antimicrobial compounds from the entire review.

 The value of a manuscript increases through value of statements. From the SAR point of view, it is useful to identify the most valuable optimizations of the molecule that lead to increased antibacterial activity, as previously recommended.

We have added a new fragment in which we have collected the most important information on the relationship between the structure and antimicrobial activity of the 1,2,4-triazole derivatives discussed.

Round 3

Reviewer 2 Report

I appreciate the authors' effort to improve and increase the value of the review.